# Prolonged Grief Disorder and Symptoms of Anxiety and Depression among Bereaved Family Caregivers in the Context of Palliative Home Care

Alberto Sardella [1],*, Alessandro Musetti [2], Pasquale Caponnetto [3,4], Maria C. Quattropani [3] and Vittorio Lenzo [3]

1   Department of Clinical and Experimental Medicine, University of Messina, 98124 Messina, Italy
2   Department of Humanities, Social Sciences and Cultural Industries, University of Parma, 43121 Parma, Italy
3   Department of Educational Sciences, University of Catania, 95124 Catania, Italy
4   Center of Excellence for the Acceleration of Harm Reduction (CoEHAR), University of Catania, 95123 Catania, Italy
*   Correspondence: asardella@unime.it

**Abstract:** *Background*: This study aimed to investigate the prevalence and the psychological comorbidity of PGD among bereaved family caregivers of palliative care cancer patients. We also examined the discriminant validity of two simple and reliable tools in correctly categorized individuals with PGD. *Methods*: A cross-sectional study was conducted with 157 bereaved participants (77.1% females, mean age = 43.50 ± 14.04 years, mean time since the loss = 3.59 years) recruited from three palliative home care services. These participants completed the Prolonged Grief Scale (PG-13) and the Hospital Anxiety and Depression Scale (HADS). *Results*: The prevalence of PGD within the sample was 4.46% (i.e., = 7/157). Participants scored higher than the cut-off on the PG-13 and the HADS-D. Symptoms of PGD were positively correlated with depression levels. The ROC curve analysis showed that the HADS-D was outstanding in categorizing individuals with prolonged grief disorder from those without PGD. A HADS-D score of ≥7.5 was able to categorize participants with a sensitivity of 0.90 and a specificity of 0.73. *Conclusions*: Overall, these results highlight the relationship between grief and depression symptoms and their exceptional discriminant validity among correctly identified individuals with PGD.

**Keywords:** clinical psychology; palliative care; caregiver; prolonged grief; loss; cancer; anxiety; depression

## 1. Introduction

The death of a loved one is generally considered among the most stressful experiences that individuals face during their lives [1]. Although most mourners return to normal functioning with a renewed sense of resilience [2], approximately 10% [3] are at risk of developing a syndrome named prolonged grief disorder (PGD), as described by Prigerson and colleagues [4]. The Diagnostic and Statistical Manual of Mental Disorders, fifth edition, text revision (DSM-5-TR) included PGD in Section 2, under the chapter entitled "trauma- and stressor-related disorders" [5]. The DSM-5-TR criteria for PGD lists the death of a loved one in at least the last 12 months, as well as the onset of a chronic grief reaction characterized by deep yearning or longing for the lost person accompanied by preoccupation with thoughts or memories. Other loss-related symptoms, including disturbance to sense of identity, intense emotional pain and emotional numbness, may also occur. On the one hand, PGD has been added to the DSM-5-TR as a new mental disorder; on the other hand, many relevant issues have arisen from bereavement research. Undoubtedly, flourishing literature has debated its existence as a distinct clinical entity [6–8]. However, since different grief assessment tools were used in the studies, it should be noted that prevalence decreased to 3.2% when considering studies using the Prolonged Grief Scale (PG-13) [4].

Some bereaved individuals appear more vulnerable to developing PGD, such as family caregivers of cancer patients in palliative care [9]. Palliative care supports family caregivers throughout a patient's illness, though increased attention after loss is necessary to prevent prolonged grief disorder. When supporting their suffering loved ones, family caregivers of patients with advanced cancer face a stressful event that may have mental health consequences even after the loss [10]. It turned out that 7%, 11%, and 5% of family caregivers had a PGD at 6, 13, and 37 months following the death of their love one, respectively [11]. Furthermore, PGD symptoms persisted for at least three years post loss. Despite some evidence, research findings on bereavement indicate that many important questions remain unresolved. It is no coincidence that bereavement support has been defined as "the forgotten child" of the palliative care family [12]. In the context of palliative care a deeper understanding of the characteristics of family caregivers who develop a prolonged grief disorder could be useful. A cross-sectional study among a sample of Chinese people has found that slightly less than 2% of the participants had a PGD [13]. Most of them were women, graduated with a high school diploma, and declared a medical or traumatic cause of death. Unfortunately, several relevant questions arise when considering informal caregivers of terminally ill cancer patients. First of all, probably due to the lack of homogeneity in sample characteristics and time for evaluation of studies, there is some heterogeneity of the PGD among bereaved caregivers. Indeed, the prevalence of PGD and, more generally, of psychological symptoms varies from 1.80% to 50% [14–16]. In order to understand these conflicting findings, longitudinal studies have indicated that the risk of PGD decreases over time [16]. In pursuing this question, however, it may be helpful to determine the key characteristics of family caregivers with a PGD diagnosis.

The COVID-19 pandemic has profoundly affected the way we deal with the loss of a loved one [17]. Available evidence has so far reported a significant impact on the mental health of the population [18–21]. There is also evidence that social distancing and other restrictive measures to reduce the spread of contagion have impacted the bereaved. At the very least, the inability to visit a loved one in the hospital and say goodbye may increase the risk of complicated grief [22]. In further characterizing grief in the time of COVID-19, Gesi and colleagues [23] emphasized that the pandemic has several aspects in common with natural disasters. Therefore, the prevalence of complicated bereavement may be increased, even if further studies are required, as the pandemic represents an unprecedented event.

Another caveat is the relation between grief symptoms and anxiety and depression. The meta-analysis data indicate higher levels of these symptoms during the COVID-19 pandemic than pre-pandemic [24]. The intensity of symptoms of prolonged grief is the strongest predictor of PGD [16]. That said, family members who lost their loved ones to cancer may also suffer symptoms of anxiety and depression. In this respect, a cross-sectional study revealed a prevalence of 48% for anxiety and 57.6% for depression [25], thus the need to carefully consider these factors (together with symptoms of PGD). Although major depressive disorder represents the most common comorbidity [5], it has been found that the comorbidity of anxiety is also usual [26]. Further research is still necessary in the context of palliative care, which implies that loss of a loved one occurs after a chronic condition, where the family members face a stressful situation [27]. Also of particular interest is the role of anxiety and depression in discriminating individuals with PGD with adequate accuracy, in terms of sensitivity and specificity. In addition to the intensity of grief symptoms, it may be helpful for clinicians to know through simple self-report instruments what symptoms can discriminate family caregivers with PGD.

Based on these premises, the primary focus of this study was to examine the prevalence of PGD, as well as the characteristics of subjects who met the diagnostic criteria. We hypothesized a low to medium prevalence of PGD. The second purpose of this research was to investigate the comorbidity of PGD with symptoms of anxiety and depression. We hypothesized a high comorbidity with depression symptoms and a low comorbidity with anxiety symptoms. The third objective of this study was to verify the discriminant validity

of the HADS-A, HADS-D, and HADS-TOT scores in people appropriately categorized with PGD. We postulated that the HADS-D scores were more discriminating than others.

## 2. Materials and Methods

### 2.1. Participants and Procedure

The present study is part of a research project entitled "Risk and protective factors for prolonged grief disorder in family caregivers of patients in palliative home care". Some of the outcomes inherent in the psychological factors underlying the ill-adapted reactions have already been presented [28]. Participants were enrolled between February 2020 and November 2021 and were family caregivers of patients with advanced cancer who were in the care of three palliative home care services in Sicily, Italy. Clinical psychologists in these departments provided information about this study to family caregivers who were interested in participating. Afterwards, a researcher approached family caregivers who met the study criteria. Participants provided written informed consent before participating in this study and completed a battery of self-report devices anonymously. Criteria for inclusion included being at least 18 years of age, being able to give informed consent, and having lost a loved one to cancer. Potential participants who had a pre-existing mental health disorder and/or were on psychotropic medications at the time of participation in this study were excluded. PGD cases were identified using the diagnostic algorithm employed by Prigerson and colleagues [4]. The search was carried out in accordance with the 1964 Helsinki Declaration and its subsequent amendments. The privacy of the participants was ensured in accordance with the European Union General Data Protection Regulation 2016/679. This study has been approved by the University of Messina's Research Ethics Committee for Psychological Research (n. 93120). A total of 159 participants agreed to take part in this study and 157 participated.

### 2.2. Measures

Participants completed a questionnaire with single-item questions on demographics (i.e., age, gender, and education) and loss information (i.e., year of bereavement, relationship with the patient, and work status).

The participants also filled out the following self-report instruments:

The Prolonged Grief Scale (PG-13) [4] is a self-report tool to assess PGD cases and symptom levels related to the DSM-5 and ICD-11 criteria [29]. In particular, a diagnostic algorithm reflecting DSM-5 and ICD-11 criteria makes it possible to identify cases of PGD. To determine the presence of prolonged grief disorder, five items reflecting the DSM-5 criteria must be satisfied. In addition to identifying individual PGD cases, the PG-13 allows the assessment of the severity of grief symptoms through the sum of all the 11 item scores, while the other two exclusively refer to the diagnostic algorithm. The eleven elements referring to the severity of grief concern cognitive, emotional, and behavioral symptoms. Each element is rated on a 5-point Likert scale. In the current study, the Italian version of the PG-13 [30], which shows adequate psychometric properties, was used. The degree of reliability of this sample was adequate, with a Cronbach's $\alpha$ of 0.88.

The Hospital Anxiety and Depression Scale (HADS) [31,32] is a self-report tool for measuring anxiety and depression over the last week. The HADS consists of 14 elements on a 4-point Likert scale, ranging from "0" to "3". All elements are grouped into two subscales for estimating anxiety (HADS-A) and depression (HADS-D), respectively. A total score reflecting general distress can be calculated by adding the two subscales together. Scores for each subscale vary from "0" to "21", with high scores indicating higher levels of anxiety and depression. Although the HADS was initially implemented to assess symptoms in outpatients with a medical condition (e.g., cancer), it is also widely used in non-clinical samples. Studies carried out over the years have shown that the HADS is a very well-known and simple instrument [33–35]. The decision to use the HADS in the present study was also reinforced by the correct psychometric properties of its Italian version [36]. The degree

of reliability for this sample was excellent, with Cronbach's α of 0.80 for the HADS-A, 0.79 for the HADS-D, and 0.86 for the HADS-TOT.

### 2.3. Statistical Analysis

The statistical analysis was carried out using IBM SPSS Statistics version 26 (IBM Corporation, Armonk, NY, USA). First of all, the data obtained from this study were verified and, subsequently, descriptive and inferential statistical analyses were performed. The one-sample *t*-test was used to compare the results of this sample with the cut-off scores established by De Luca and colleagues for the PG-13 [30] and by Iani and colleagues for the HADS [36]. The relationships between the PG-13 and the HADS was established using Pearson product–moment correlation coefficients. The Receiver Operating Characteristic (ROC) curve was developed to evaluate the performance of the HADS-A, the HADS-D, and the HADS-TOT in categorizing individuals with prolonged grief disorder. The overall accuracy of the instruments was evaluated in the area under the ROC curve (AUC). This indicates the probability of a respondent being correctly assigned to the appropriate group. An AUC value between 0.7 and 0.8 is considered acceptable, while between 0.8 and 0.9 is deemed excellent, and more than 0.9 is deemed outstanding [37]. The Youden Index [38] was also calculated to determine the best cut-off value for the HADS-A, the HADS-D, and the HADS-TOT.

## 3. Results

### 3.1. Characteristics of the Sample

Table 1 shows the sample demographic and losses. Overall, 157 subjects were included in this study. Most of the participants were female (*n* = 121; 77.1%) and had a high school diploma (n = 66; 42%), while the mean age in years was 43.50 ± 14.04 (range 18–81). In terms of the loss characteristics, most of the participants were sons or daughters of the loved one (n = 82; 52.3%) and were the principal caregiver during the time of palliative care (n = 90; 57.3%). The mean time since the death of the loved one was 3.59 years (SD = 4.92, range = 1–28). Lastly, approximately half of the participants declared working before the loss (n = 81; 51.6%). This prevalence increased after the loss (n = 94; 59.9%).

**Table 1.** Characteristics of the sample.

| Characteristics | n (%) | M | SD |
|---|---|---|---|
| Age (in years) | | 43.50 | 14.04 |
| Gender | | | |
| Male | 36 (22.9) | | |
| Female | 121 (77.1) | | |
| Education | | | |
| Primary or middle school diploma | 34 (21.7) | | |
| High school diploma | 66 (42) | | |
| Graduate | 57 (36.3) | | |
| Relation with the deceased loved one | | | |
| Son or daughter | 82 (52.3) | | |
| Nephew | 32 (20.4) | | |
| Spouse | 15 (9.5) | | |
| Other (for example, brother-in-law) | 28 (17.8) | | |
| Main caregiver | | | |
| Yes | 90 (57.3) | | |
| No | 67 (42.7) | | |
| Work before the loss | | | |
| Yes | 76 (48.4) | | |
| No | 81 (51.6) | | |
| Work after the loss | | | |
| Yes | 63 (40.1) | | |
| No | 94 (59.9) | | |
| Time since the loss (years) | | 3.59 | 4.92 |

### 3.2. Prevalence of PGD, Depression, and Anxiety

The prevalence of prolonged grief disorder among family caregivers was 4.46% (n = 7). Figure 1 illustrates the average of the PGD symptoms for the sample. The longing or yearning mean was higher than the other PGD symptoms (*M* = 3.61, *SD* = 1.28), while difficulties moving on was the lowest (*M* = 1.59, *SD* = 0.92).

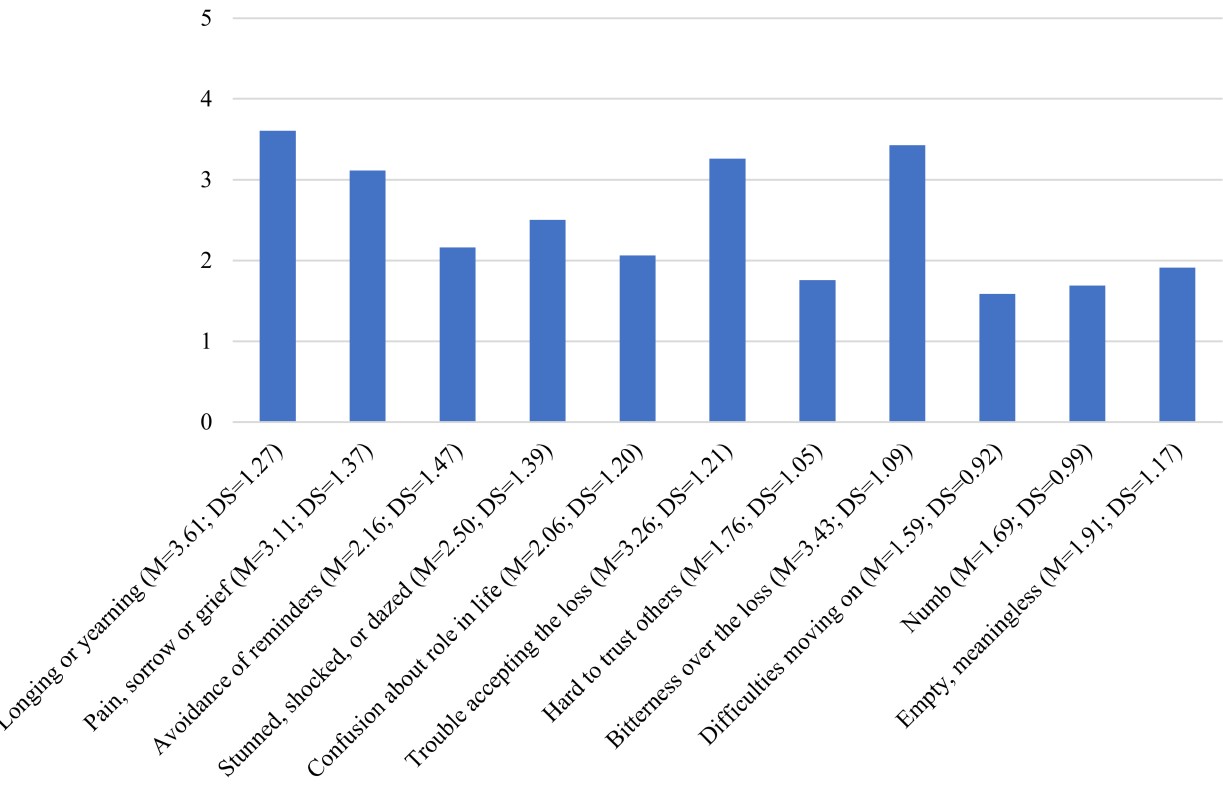

**Figure 1.** Mean scores of PGD symptoms in bereaved caregivers following the PG-13.

Table 2 displays the characteristics of the seven participants who met the criteria for the diagnosis of prolonged grief disorder. All the participants with PGD were female and the main caregiver of the deceased loved one. Most of them were daughters and did not work before the loss, while the age ranged from 24 to 75 years, and the time since the loss was between one and two years. As expected, the participants with PGD scored above the cut-off for the PG-13, the HADS-A, the HADS-D, and the HADS-TOT. Specifically, most of the HADS mean scores were 2-fold higher than the scores obtained by healthy subjects [36]. Likewise, the participants with PGD had a higher severity of symptoms, similar to what was obtained by De Luca and colleagues [30].

**Table 2.** The characteristics of seven participants with PGD.

| I.D. | Age | Gender | Education | Kinship | Main Caregiver | Work before the Loss | Work after the Loss | Time | PG-13 | HADS-A | HADS-D | HADS-TOT |
|------|-----|--------|-----------|---------|----------------|----------------------|---------------------|------|-------|--------|--------|----------|
| 1 | 73 | F | Middle school | Spouse | Yes | No | No | 1 | 39 | 14 | 13 | 27 |
| 2 | 51 | F | High school | Daughter | Yes | No | Yes | 1 | 42 | 11 | 8 | 19 |
| 3 | 24 | F | High school | Daughter | Yes | Yes | Yes | 1 | 46 | 7 | 12 | 19 |
| 4 | 35 | F | Graduate | Daughter | Yes | No | No | 1 | 53 | 14 | 14 | 28 |
| 5 | 75 | F | Middle school | Spouse | No | No | No | 2 | 51 | 19 | 18 | 37 |
| 6 | 49 | F | Middle school | Daughter | Yes | No | Yes | 2 | 48 | 13 | 15 | 28 |
| 7 | 57 | F | High school | Daughter | Yes | Yes | Yes | 2 | 40 | 7 | 7 | 14 |

Note: Age = age of the subject; F = female; Middle school = primary or middle school diploma; High school = high school diploma; Kinship = relation with the deceased loved one; Time = time since the loss in years; PG-13 = PG-13 scores of the severity/intensity of the symptoms; HADS-A = HADS-Anxiety scores; HADS-D = HADS-Depression scores; HADS-T = HADS general distress scores.

### 3.3. Comorbidity of PGD, Depression, and Anxiety

Table 3 shows the descriptive statistics and results of correlation analyses. The mean value of the PG-13 ($M$ = 27.08, $SD$ = 8.84) was significantly higher than the mean score indicated by the Italian validation study of 23.4 [30], $t(156)$ = 5.21, $p$ < 0.001, while the scale score range was very similar. The HADS-A and HADS-TOT mean scores were near the mean scores referring to healthy subjects [36] and no significant differences were found [respectively, $t(156)$ = −0.78, $p$ = 0.44 for the HADS-A and $t(156)$ = 1.05, $p$ = 0.29 for the HADS-TOT]. The results also showed that the mean value of the HADS-D ($M$ = 6.22, $SD$ = 3.56) was significantly higher than the healthy population indicated by Iani and colleagues [36], $t(156)$ = 2.90, $p$ = 0.004. The correlation coefficients between the PG-13 and the HADS-A, the HADS-D, and the HADS-TOT were 0.37, 0.54, and 0.50, respectively ($p$ < 0.01).

**Table 3.** Descriptive and correlational analyses.

| Variable | Min | Max | M | SD | 1. 1. PG-13 | 2. HADS-A | 3. HADS-D |
|---|---|---|---|---|---|---|---|
| 1. PG-13 | 11 | 53 | 27.08 | 8.84 | | | |
| 2. HADS-A | 0 | 19 | 7.34 | 4.11 | 0.37 ** | | |
| 3. HADS-D | 0 | 18 | 6.22 | 3.56 | 0.54 ** | 0.61 ** | |
| 4. HADS-TOT | 1 | 37 | 13.57 | 6.74 | 0.50 ** | 0.91 ** | 0.89 ** |

Note: ** $p$ < 0.01.

### 3.4. Discriminant Validity of HADS-A, HADS-D, and HADS-TOT Scores

As shown in Figure 2, the ROC curve analysis indicated that the HADS-D was outstanding in categorizing individuals with prolonged grief disorder from those without PGD [AUC = 0.92, SE = 0.04, $p$ < 0.001, CI = 0.85, 0.99]. A score of ≥7.5 on the HADS-D (Youden index = 0.63) categorized participants with a sensitivity of 0.90 (90% of all participants with prolonged grief disorder were correctly categorized) and a specificity of 0.73 (27% of participants were incorrectly categorized).

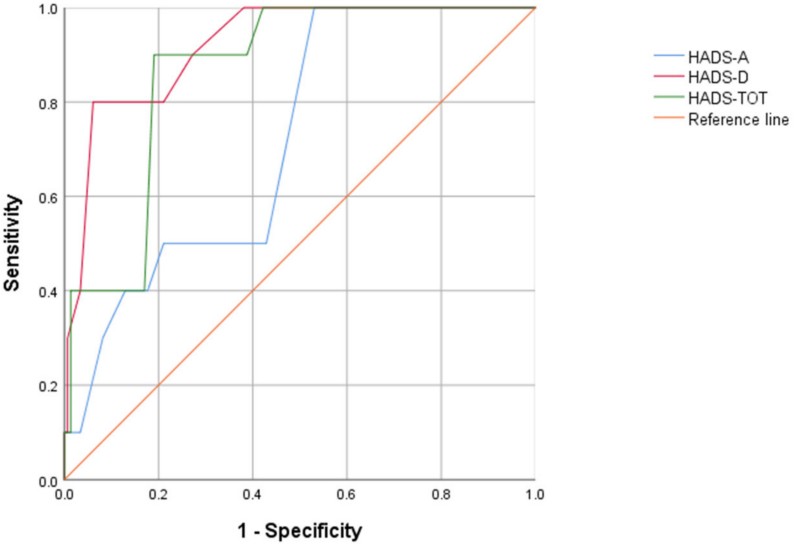

**Figure 2.** ROC curve graph for the HADS-A, the HADS-D, and the HADS-TOT to discriminate between individuals with and without prolonged grief disorder.

When considering anxiety, the ROC curve analysis showed that the HADS-A was only acceptable in correctly identifying individuals with prolonged grief disorder [AUC = 0.72, SE = 0.07, $p$ < 0.05, CI = 0.58, 0.86]. A score of ≥10.5 on the HADS-A (Youden index = 0.29) categorized participants with a sensitivity of 0.50 (50% of all individuals with prolonged

grief disorder were correctly categorized) and a specificity of 0.79 (21% of subjects were incorrectly categorized).

Lastly, considering general distress, a ROC curve procedure showed that the HADS-TOT could classify participants with prolonged grief disorder [AUC = 0.87, SE = 0.04, $p < 0.001$, CI = 0.78, 0.95]. A score of $\geq$18.5 on the HADS-TOT (Youden index = 0.71) categorized participants with a sensitivity of 0.90 (90% of all participants with prolonged grief disorder were correctly categorized) and a specificity of 0.81% (19% of participants were incorrectly classified).

## 4. Discussion

To date, bereavement research in the context of palliative home care has left several unresolved issues. This study sought to investigate the prevalence and characteristics of PGD among family caregivers, its comorbidity with anxiety and depression, as well as the discriminant validity for the PGD diagnosis of some of the most used self-report instruments. In particular, we first examined the prevalence of PGD diagnosis in our sample. Our findings revealed that approximately 4 percent of the family caregivers had a PGD diagnosis. This result appears coherent with what have found in other studies. Despite the high heterogeneity of the research in the field, indeed meta-analytic evidence converged in indicating that the pooled prevalence of PGD is 3.2% when data were gathered with the Prolonged Grief Scale [3]. Still, the high degree of heterogeneity between studies [14–16] should be taken into account in future research. However, as an explanation of that conclusion, the average time since loss should also be taken into consideration. The fact that participants were assessed an average of 3.6 years after their loss may explain the low prevalence of PGD that we found. For instance, a longitudinal study of 301 bereaved cancer caregivers found that the prevalence of PGD decreased from six months to three years [11]. Specifically, they reported a 5% PGD rate among the caregivers, consistent with our findings. The fact that the prevalence of symptoms tends to decrease over time [3] supports the view that most individuals with potentially traumatic experiences showed resilience [39].

It is also important to understand sociodemographic and loss characteristics as well as anxiety and depression levels. This should be seen within the context of palliative care as a means of enhancing the role of prevention. As our results suggest, people with PGD were daughters who mainly assisted their loved ones and had no job, while their age was greatly variable. Indeed, our results seem to resemble that of the literature. For example, He and colleagues [13] discovered that most of the people with PGD were women. However, the interpretation of this finding should be interpreted with caution, as both samples consisted primarily of women. Furthermore, this finding may also be related to the adopted variable-oriented research approach [40].

The second goal of this study was to explore the psychological morbidities of the symptoms of grief among family caregivers. In a grieving state characterized by the risk of developing a mental disorder such as PGD, it is not surprising that other psychological symptoms may develop. The identification of the potential comorbidities in the grief response is at the forefront of most clinicians and researchers. The literature has well established that symptoms of depression and anxiety are very common after loss [41,42].

That many family caregivers suffer from these symptoms is not surprising in light of the well-established comorbidities. It is worth underlining, however, that several studies have reported increased anxiety and depression levels during the COVID-19 pandemic [18–21]. The literature also converged in indicating that restrictive measures following the pandemic, along with the fear of contagion, might harm bereaved people struggling with grief [22,23]. Although disentangling the consequences of an unprecedented pandemic on mental health is challenging, high symptom levels observed among bereaved individuals during the pandemic, especially in countries most affected by the pandemic, should be considered with caution. Hence, bearing in mind these considerations, we found that family caregivers scored higher on the severity of the symptoms of grief than the cut-off

detected by De Luca and colleagues [30]. The COVID-19 pandemic may have contributed to this finding. Likewise, the depression scores of our sample were higher than that of the healthy subjects [36], while the anxiety scores were approximately similar. As would be expected, the relationship between the severity of grief symptoms and depression was the strongest, supporting the relevant comorbidity. This is, after all, additional evidence that that depressive disorders are the most frequent comorbidities [5]. Because the common manifestations of such mental disorders are visible, differential diagnosis is both difficult and paramount. Clinicians and researchers have consistently drawn attention to how the symptoms of grief are distinct phenomena. On the one hand, it has been argued that seeking proximity to the dead person is keenly related to complicated grief; on the other hand, anhedonia and self-critical feelings seem to be more common in depression [43]. In any case, it is necessary to consider the sensitivity and specificity of two of the most used instruments in the field of bereavement research.

In this vein, the last aim of this study was to examine the discriminant validity of the HADS's subscale as well as its total score in precisely diagnosed family caregivers with PGD, according to the DSM criteria. Since PGD was included in the DSM-5-TR as a new mental disorder [5], there is an increasing interest in reliable and simple instruments. It is worth pointing out that the PG-13 items do not grasp all criteria for PGD [44]. By the same token, we investigated the discriminant validity of a very simple and popular tool such as the HADS, and more generally, its HADS-D and HADS-A subscales, as well as the HADS-TOT. Despite the fact that it was originally proposed to measure depression and anxiety among outpatients in hospitals, several studies conducted over the years have dwelled on its use for screening as well as for case finding [33–35]. Yet, a study that tested the factorial structure in a large Italian sample confirmed the use of the bifactor model, with a general psychological distress factor and two factors with anxiety and depression [36]. Accordingly, the HADS may represent a valid instrument for identifying PGD cases. Our findings revealed that the HADS-D was outstanding in categorizing individuals with prolonged grief disorder from those without PGD. Specifically, 90% of participants with prolonged grief disorder were correctly categorized. After the statistical analysis, however, not all of the study findings may be considered adequate. As for the HADS-A, indeed analysis of the ROC curve found that only 50% of the family caregivers were correctly identified as PGD cases, while 21% of them were incorrectly diagnosed. Undoubtedly, our findings did not support the use of the HADS-A for correctly identifying individuals with PGD. Given this conclusion, the interpretation may be related to the fact that the HADS-A essentially assesses the psychic manifestations of anxiety neurosis [31]. Instead, relative to the total HADS score, indicating general psychological distress, the results of the ROC curve procedure suggested that approximately 20% of the family caregivers were improperly categorized as PGD cases, though the sensitivity remained excellent.

There are some limitations to this study that should be addressed in forthcoming research. First, the use of a cross-sectional design failed to verify any change in observed variables over time. Longitudinal studies would shed light on the fluctuating levels of grief symptoms, anxiety and depression among family caregivers of deceased patients in the context of palliative care. Second, the oversampling of some characteristics of the sample such as the female gender may have affected the results and, subsequently, it may be not possible to generalize our findings to other family caregivers who lost a loved one. On the other hand, there are some characteristics (i.e., the female sex) that can be considered emblematic of caregivers who assist their loved ones with terminal cancer. Third, the data from this study have been gathered with the PG-13 that corresponds to the DSM-5 criteria. Although Prigerson and colleagues [45] have proposed a revised version of the scale grounded on the DSM-5-TR criteria for PGD, to date, it is not available in the Italian context. Fourth, since the data were collected in 2020 and 2021, the psychological impact of the restrictive measures during the COVID-19 pandemic on the general population might have influenced the reported symptoms of anxiety and depression as well as PGD. Further post-pandemic research would help clarify if the climate of high infection rates has

influenced the results. Finally, other variables may have influenced caregivers' levels of anxiety and depression, particularly in older adults [46]. Future research should address the role of these variables in influencing outcomes.

## 5. Conclusions

In summary, our findings indicated that the prevalence of PGD in our sample was approximately 5%. The severity of grief symptoms was positively related to depression and, to a lesser extent, with anxiety. The HADS-D was the best option for identifying family caregivers with PGD. Our findings indicated that 90% of the PGD cases were correctly identified (no false negative results), while only 27% of the participants were incorrectly classified (few positive results). Our findings suggest that bereaved family caregivers should be more sensitive to PGD risk, in a field closely related to loss such as home palliative care for cancer patients, and this would be a promising approach to psychological prevention. From this perspective, our findings can be used to refine targeted intervention strategies for clinicians and researchers in the field.

**Author Contributions:** Conceptualization, V.L. and M.C.Q.; methodology, V.L., A.M. and A.S.; formal analysis, A.S. and V.L.; investigation, A.S.; data curation, A.S.; writing—original draft preparation, A.S. and V.L.; writing—review and editing, P.C., V.L. and A.M.; supervision, V.L. and M.C.Q.; project administration, V.L. and M.C.Q. All authors have read and agreed to the published version of the manuscript.

**Funding:** This research received no external funding.

**Institutional Review Board Statement:** This study was conducted in accordance with the Declaration of Helsinki, and approved by the Research Ethics Committee for Psychological Research of the University of Messina (n. 93120).

**Informed Consent Statement:** Informed consent was obtained from all subjects involved in this study.

**Data Availability Statement:** The raw data supporting the conclusions of this article will be made available by the authors, without undue reservation.

**Acknowledgments:** We thank Cristina Faraone for her help with this project.

**Conflicts of Interest:** The authors declare no conflict of interest.

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
