# Peer review of "Prolonged Grief Disorder and Symptoms of Anxiety and Depression among Bereaved Family Caregivers in the Context of Palliative Home Care"

_ejihpe, doi:10.3390/ejihpe13020037_

Round 1
Reviewer 1 Report
The authors have conducted a broadly competent study of the incidence of PGD in a home-based palliative care context, drawing on a sizeable sample of family caregivers who completed three well validated assessments of prolonged grief, depression and anxiety. For the most part their rationale for the study was cogent and their interpretations defensible. However, 5 aspects of the report require revision and/or acknowledgement as limitations:
1. With the inclusion of PGD in the DSM 5-TR early this year, the previous formulation of PCBD becomes obsolete, and should be dropped as a context for proposing the study or interpreting its results. Instead, rely on the DSM 5-TR diagnosis throughout.
2. All data for this study of bereavement were collected in 2020 and 2021, as Italy and the world were reeling from the first and second waves of Covid-19. This context is essential to acknowledge in the Introduction, as a great deal of well-replicated research documents the heavy toll of grief from any cause during the pandemic with its various restrictions on gatherings, social support, funerals, etc., all of which have been found to be associated with dysfunctional grief, depression and anxiety. The latter two are especially likely to be heightened as a function of the direct and indirect stressors of the pandemic, and given the frailty of the patients whose family members were studied, it is entirely likely that many of them died of Covid-19. This fact also makes it essential to consider restrictions on generalization of the study in the Discussion, as the elevations in psychiatric symptoms observed may not be a function of the loss as much as the climate of high infection rates throughout the study period, especially in Italy. For an entry into this literature, simply search for "pandemic grief" in the literature.
3. A significant weakness of the study is that participants were 3.6 years, on average, after their loss. The study would have been more informative about the incidence of PGD at the diagnostic threshold of 12-24 months; it is unsurprising that the incidence observed of about 5% was half that in many studies for this reason alone. This factor of time since loss must be acknowledged in the abstract and Discussion to contextualize the findings.
4. Although it may be informative in some sense to report that the PG-13 was "outstanding" in its diagnostic sensitivity and specificity, this finding is in many ways tautological, as the PG-13 was used as the criterion for diagnosis (i.e. extreme scores on selected items to establish "caseness") as well as predictor (adding a few more items to this set of the same items, whose extreme scores "predict" PGD-- based on a subset of the same items! In this sense the study is more like a study of the internal consistency of the PG-13 than it is a study of its predictive efficiency, which would seem to require an external measure or assessment, such as an interview-based determination of diagnosis by clinicians who were blind to the PG-13 scores). Of course, the prediction of PGD by the HADS does not suffer from this confound, so this latter analysis has value. But the value of the PG-13 to PG-13 prediction strikes me as more dubious.
5. Finally, although the study was generally clearly written, there were a few dozen errors or curious phrasings in the use of English, suggesting the value of thorough editing for language by professional proofreaders who have first-language competency in written English. Many such services are available.
Author Response
Comments from Reviewer #1:
[The authors have conducted a broadly competent study of the incidence of PGD in a home-based palliative care context, drawing on a sizeable sample of family caregivers who completed three well validated assessments of prolonged grief, depression and anxiety. For the most part their rationale for the study was cogent and their interpretations defensible. However, 5 aspects of the report require revision and/or acknowledgement as limitations:]
Reply: We thank the Reviewer for the careful reading of the manuscript and constructive remarks. We have sagely taken the comments on board to improve the manuscript. Please see above a detailed point-by-point response to his/her comments.
[1. With the inclusion of PGD in the DSM 5-TR early this year, the previous formulation of PCBD becomes obsolete, and should be dropped as a context for proposing the study or interpreting its results. Instead, rely on the DSM 5-TR diagnosis throughout.]
Reply: Thank you for this precious comment. We do agree with the Reviewer insofar as the DSM-5-TR moves the Prolonged Grief Disorder (PGD) from the “Conditions for Further Study” section to the “trauma- and stressor-related disorders” chapter. Indeed, our research project started when the DSM-5-TR had not yet been published. In the revised version of the manuscript, we have referred to the latter version. Consequently, we have modified the Introduction and Discussion sections and, specifically, the part inherent to the description of the PGD. We have also acknowledged as a further limitation of this study that data have been gathered with the PG-13 corresponding to the DSM-5 criteria.
[2. All data for this study of bereavement were collected in 2020 and 2021, as Italy and the world were reeling from the first and second waves of Covid-19. This context is essential to acknowledge in the Introduction, as a great deal of well-replicated research documents the heavy toll of grief from any cause during the pandemic with its various restrictions on gatherings, social support, funerals, etc., all of which have been found to be associated with dysfunctional grief, depression and anxiety. The latter two are especially likely to be heightened as a function of the direct and indirect stressors of the pandemic, and given the frailty of the patients whose family members were studied, it is entirely likely that many of them died of Covid-19. This fact also makes it essential to consider restrictions on generalization of the study in the Discussion, as the elevations in psychiatric symptoms observed may not be a function of the loss as much as the climate of high infection rates throughout the study period, especially in Italy. For an entry into this literature, simply search for "pandemic grief" in the literature.]
Reply: Many thanks for your comment insofar as touching on a central tenet of the manuscript. Starting from your hint on “pandemic grief” we have deepened how the pandemic has influenced bereaved people facing grief. Accordingly, we have incorporated this issue in the revised version of the manuscript. We agree that the generalization of our findings may be limited by the pandemic. In the revised version of the manuscript, indeed, we have modified the Introduction and the Discussion section. Indeed, we have added a part concerning the impact of the Covid-19 pandemic on mental health and, more specifically, on the grief among bereaved people. In the Discussion, we have reported our findings on the psychological morbidities of the symptoms of grief among family caregivers in light of the role of the Covid-19 pandemic. Finally, we have reported as a limitation the fact that the Covid-19 pandemic may have influenced the results we obtained.
[3. A significant weakness of the study is that participants were 3.6 years, on average, after their loss. The study would have been more informative about the incidence of PGD at the diagnostic threshold of 12-24 months; it is unsurprising that the incidence observed of about 5% was half that in many studies for this reason alone. This factor of time since loss must be acknowledged in the abstract and Discussion to contextualize the findings.]
Reply: Many thanks to the Reviewer for this valuable hint insofar as it allows us to improve the description of the limitations of our study. Undoubtedly, the fact that participants were 3.6 years after the death of their loved one represents a relevant issue for understanding the findings. In the revised version of the manuscript, we have added time since the loss in the abstract. Likewise, we have reported this information in the Discussion section, together with a deeper consideration of its implication for our findings.
[4. Although it may be informative in some sense to report that the PG-13 was "outstanding" in its diagnostic sensitivity and specificity, this finding is in many ways tautological, as the PG-13 was used as the criterion for diagnosis (i.e. extreme scores on selected items to establish "caseness") as well as predictor (adding a few more items to this set of the same items, whose extreme scores "predict" PGD-- based on a subset of the same items!. In this sense, the study is more like a study of the internal consistency of the PG-13 than it is a study of its predictive efficiency, which would seem to require an external measure or assessment, such as an interview-based determination of diagnosis by clinicians who were blind to the PG-13 scores. Of course, the prediction of PGD by the HADS does not suffer from this confound, so this latter analysis has value. But the value of the PG-13 to PG-13 prediction strikes me as more dubious.]
Reply: Many thanks to the Reviewer for this valuable hint insofar as it allows us to improve the description of the use of PG-13. Admittedly, using the PG-13 values for PG-13 prediction can leave readers a little perplexed. Nevertheless, on one hand, we used the PG-13 diagnostic algorithm (Prigerson et al., 2009) for identifying PGD cases insofar as the output is in terms of “yes/no”. On the other hand, the PG-13 also gives us the severity of symptoms through the sum of eleven items rated on a 5-point Likert scale. The latter was used to predict the former. To make clearer this choice, we have improved the PG-13 description in the “2.2. Measures” paragraph of the revised manuscript. Moreover, in the “2.1. Participants and Procedure” paragraph, we have deepened how the diagnosis of PGD was determined.
[5. Finally, although the study was generally clearly written, there were a few dozen errors or curious phrasings in the use of English, suggesting the value of thorough editing for language by professional proofreaders who have first-language competency in written English. Many such services are available.]
Reply: Many thanks for this comment allowing to improve the clarity of the text for the journal readers. As kindly requested, a professional proofreader who has first-language competency in written English has edited our manuscript.
Reviewer 2 Report
This report is a study to determine the prevalence of prolonged grief disorders and to test the discriminant validity of the PG-13, HADS-A, HADS-D, and HADS-TOT in bereaved family caregivers of palliative care cancer patients.
Although this is a valuable report, several questions of scientific validity need to be addressed.
1. Please specify the process by whom and how the diagnosis of plolonged grief disorder was determined. If you tested the discriminant validity of the PG-13, you did not use the PG-13 as a diagnostic tool, did you?
2. It is necessary to clearly state the rationale for why we decided to use the HADS, a screening scale for anxiety and depression, to discriminate between plolonged grief disorder.
3. If the validity of the existing scale is to be verified, the already existing cutoff values should be used, and setting cutoff values in the present data may overestimate performance and is not appropriate.
4. Although the mean scores of PGD symptoms are shown in Figure 1, they are values for each symptom, and connecting them with a line is not the correct way to present the figure. In addition, it would be better if an indicator of variability is added in addition to the mean scores.
Author Response
Comments from Reviewer #2:
[This report is a study to determine the prevalence of prolonged grief disorders and to test the discriminant validity of the PG-13, HADS-A, HADS-D, and HADS-TOT in bereaved family caregivers of palliative care cancer patients.
Although this is a valuable report, several questions of scientific validity need to be addressed.
Reply: We thank the Reviewer for the careful reading of the manuscript and for precious remarks about the manuscript. We have wisely taken the comments on board to improve the manuscript. Please see above a detailed point-by-point response to his/her comments.
[1. Please specify the process by whom and how the diagnosis of prolonged grief disorder was determined. If you tested the discriminant validity of the PG-13, you did not use the PG-13 as a diagnostic tool, did you?]
Reply: Many thanks for the opportunity to clarify this point regarding the use of PG-13. Indeed, PG-13 allows obtaining two kinds of scores. To identify the PGD cases, we used the PG-13 diagnostic algorithm implemented by Prigerson and colleagues (2009). This algorithm gives an output for diagnosis in terms of “yes/no”. Nevertheless, the PG-13 also allows to measure the severity of symptoms through the sum of eleven items rated on a 5-point Likert scale. Together with the HADS scores, only the severity of grief symptoms scores was used to investigate the discriminant validity. In the revised version of the manuscript, we have improved the PG-13 description in the “2.2. Measures” paragraph. We have also described in the “2.1. Participants and Procedure” paragraph how the diagnosis of PGD was determined.
[2. It is necessary to clearly state the rationale for why we decided to use the HADS, a screening scale for anxiety and depression, to discriminate between prolonged grief disorder.]
Reply: Many thanks to the Reviewer for this comment insofar as it allows us to explain more in deep the rationale underlying the use of the HADS. Many instruments have been used in bereavement research for measuring anxiety and depression, including the Beck Depression Inventory (BDI), the Beck Anxiety Inventory (BAI), the depression and anxiety scales of the Symptom Check List-90-R (SCL-90-R), the Center for Epidemiologic Studies Depression Scale (CES-D), and the Hospital Anxiety and Depression Scale (HADS). Among these, we chose the latter on the grounds that it is a very popular and simple self-report scale. Although it was originally developed to assess depression and anxiety among outpatients in nonpsychiatric hospital clinics, several studies over the years have drawn attention to its use for screening as well as for case-finding [(for example, Brennan and colleagues (2010)]. By doing so, we have taken into account its psychometric properties. In this light, Iani and colleagues (2014) have analyzed the factorial structure HADS in a large community sample in Italy, confirming the bifactor model, with a general psychological distress factor and two orthogonal group factors with anxiety and depression. Accordingly, we decided to include the HADS in our study. In the revised version of the manuscript, we have better explained the rationale for the use of the HADS among the sample of bereaved people involved in the present study. In this vein, we have modified the description of the instrument (paragraph “2.2. Measures”) as well as the Discussion section, when we have reported findings related to the last aim of the study.
[3. If the validity of the existing scale is to be verified, the already existing cutoff values should be used, and setting cutoff values in the present data may overestimate performance and is not appropriate.]
Reply: Many thanks for this precious indication. We referred to the cut-off indicated by De Luca and colleagues (2015) for PG-13 and by Iani and colleagues (2014) for HADS. To compare our results with the existing cut-off values, a series of one-sample t-tests were performed. We have added the results we obtained in the “3.2. Comorbidity of PGD, depression, and anxiety” paragraph. These results confirmed what we have found for descriptive analysis. Also, we added a description of this further analysis in the “2.3. Statistical Analysis” paragraph.
[4. Although the mean scores of PGD symptoms are shown in Figure 1, they are values for each symptom, and connecting them with a line is not the correct way to present the figure. In addition, it would be better if an indicator of variability is added in addition to the mean scores.]
Reply: Thank you for your comment. In the revised version of the manuscript, we have modified Figure 1 from “a scatter chart with connecting lines” to “a clustered chart”. We have also added the standard deviation for each mean score.
Round 2
Reviewer 1 Report
Professional editing for use of English required.
Author Response
Reviewer 1: Professional editing for use of English required.
Authors’ reply: We kindly confirm that a mother tongue consultant had already accurately revised the language.
Reviewer 2 Report
I think it has been corrected carefully and improved a lot.
On the other hand, it was methodologically incorrect to diagnose prolonged grief disorder with the PG-13 diagnostic algorithm and validated the discriminant validity with the PG-13 score. It is not surprising that the discriminative power was outstanding since it is the same scale. Therefore, the results regarding the PG-13 score should be removed.
Author Response
REVIEWER 2: I think it has been corrected carefully and improved a lot. On the other hand, it was methodologically incorrect to diagnose prolonged grief disorder with the PG-13 diagnostic algorithm and validated the discriminant validity with the PG-13 score. It is not surprising that the discriminative power was outstanding since it is the same scale. Therefore, the results regarding the PG-13 score should be removed.
Authors’ reply: The results pointed by the Reviewer have been deleted from the revised manuscript, in order to address the reviewer’s point of view. Anyway, we would kindly like to say that PG-13 provides both an algorithm for diagnosing complicated grief disorder and a symptom severity score. We used the PG-13 score to predict the categorical diagnosis. However, to avoid issues and to welcome the reviewer’s comment, we have removed this part on symptom severity to predict grief diagnoses, maintaining only the HADS score.
Round 3
Reviewer 2 Report
Thank you for my acceptance and appropriate correction. Of course, I am aware that PG-13 provides both a diagnostic algorithm and a symptom severity score for complicated grief syndrome. However, I do not understand the scientific significance of using the same scale severity score to predict a category diagnosis identified by the PG-13 diagnostic algorithm.